# The Effect of Surface Wettability on Viscoelastic Droplet Dynamics under Electric Fields

**DOI:** 10.3390/mi13040580

**Published:** 2022-04-07

**Authors:** Bo Sen Wei, Sang Woo Joo

**Affiliations:** School of Mechanical Engineering, Yeungnam University, Gyeongsan 38541, Korea; wbs1996919@naver.com

**Keywords:** droplet deformation, viscoelasticity, wettable surface, dielectric field

## Abstract

The effects of surface wettability and viscoelasticity on the dynamics of liquid droplets under an electric field are studied experimentally. A needle-plate electrode system is used as the power source to polarize a dielectric plate by the corona discharge emitted at the needle electrode, creating a new type of steerable electric field realized. The dynamics of droplets between the dielectric plate and a conductive substrate include three different phenomena: equilibrium to a stationary shape on substrates with higher wettability, deformation to form a bridge between the top acrylic plate and take-off on the substrates with lower wettability. Viscoelastic droplets differ from water in the liquid bridge and takeoff phenomena in that thin liquid filaments appear in viscoelastic droplets, not observed for Newtonian droplets. The equilibrated droplet exhibits more pronounced heights for Newtonian droplets compared to viscoelastic droplets, with a decrease in height with the increase in the concentration of the elastic constituent in the aqueous solution. In the take-off phenomenon, the time required for the droplet to contact the upper plate decreases with the concentration of the elastic constituent increases. It is also found that the critical voltage required for the take-off phenomenon to occur decreases as the elasticity increases.

## 1. Introduction

In recent years, applications of new microfluidic technologies have grown tremendously, especially in biological, chemical, optoelectronic tweezers technology [1,2], biomedical, and other thermofluidic operations. Controlling microfluidic operations with better efficiency and accuracy has become a key part in recent years [3,4]. An important driving mechanism of fluids in microfluidics is by use of the electricity, as is common in many processes, such as electrohydrodynamic (EHD) atomization [5], electrospinning [6], inkjet printing [7], dielectrophoresis [8,9,10,11], electrowetting [12], polymer patterning preparation [13], and electrostatic spraying [14], among others. As a representative example, the deformation of thin films or droplets caused by an applied electric field is of scientific interest and a practical importance that has been studied experimentally and theoretically for many decades. The deformation of droplets before reaching the Rayleigh limit [15,16,17,18] and the droplet ejection after rupture [19,20] both have been enticing subjects of investigations. Depending on the physical properties of droplets, phenomena such as jetting and electrospraying can occur [21,22]. When a droplet is subject to an applied electric field, it undergoes deformation due to the electrostatic stresses exerted on the interface.

Swan [23] discovered the charge-induced stress in resins and viscous compounds of resins and oils. Cheng and Miksis [24] investigated the shape and the stability of droplets on conducting planes in electric fields. Basaran and Scriven [25] studied the relative importance of the electricity and the gravity compared to the surface tension. Wohlhuter and Basaran [26] found that, regardless of the ratio of the permittivity of the droplet to that of the surrounding fluid κ, the droplet shape exhibits a conical tip as its deformation develops. Three types of behaviors are found, depending on the value of κ. For κ<20.25, the droplet deformation grows without bound as the field strength rises. On the other hand, for 21.75≥κ>20.25, families of equilibrium droplet shapes become unstable at turning points with respect to the field strength. For κ>21.75, results predict that droplet deformations exhibit hysteresis. Reznik et al. [27] evolution of small droplets attached to a conducting surface and subjected to a relatively strong electric field has been investigated experimentally and numerically. Three different droplet-shape evolution scenarios are distinguished based on a numerical solution of the Stokes equation for perfectly conducting droplets. In a sufficiently weak (subcritical) electric field, the droplet is stretched by Maxwell electrical stress, and acquires a steady-state shape, where equilibrium is achieved with the action of the surface tension. In a stronger (supercritical) electric field, the Maxwell stress overcomes the surface tension, and for static (initial) contact angle of the droplet with the conducting electrode $\alpha_{s}<0.8\pi \;$ an ejection starts from the tip of the droplet. In this case, the base of the jet acquires a quasi-stable, nearly conical shape with a vertical half angle β≤30°, which is significantly smaller than the Taylor cone ($\beta_{T}=49.3^{{}^{\circ}}$). Finally, in a supercritical electric field acting on a droplet with a contact angle in the range of $0.8\pi <\alpha_{s}<\pi$, there is no ejection but the entire droplet jumps off, as reported by Mugele and Baret [28]. Corson et al. [29] investigated the electric field-induced deformation of a nearly hemispherical conducting droplet theoretically and temporally. Tsakonas et al. [30] studied the electric field-induced deformation of hemispherical sessile droplets of ionic liquid. Sessile droplets of an ionic liquid with contact angles close to 90° were subjected to an electric field E=V/h inside a capacitor with plate separation h and potential difference V. For small field induced deformations of the droplet shapes the change in maximum droplet height $\Delta H=H\left(E\right)-H\left(0\right)$ was found to be virtually independent of the plate separation provided that $h>3H\left(0\right)$.

In the study of droplet behavior on electric field-controlled superhydrophobic surfaces, A. Glière [31] studied the complex lift-off process caused by the competition between gravity, electricity and capillary forces. The results of B. Traipattanakul [32] show that with the increase in the plate electrode gap width, both the voltage threshold and the electric field threshold increase, while the droplet charge decreases. Christos Stamatopoulos [33] manipulated the droplet discharge by changing the wettability, and illustrated the difference between the strength of the applied electric field and the deformed shape of the droplet. Arshia Merdasi [34] analyzed electrowetting-induced droplet hopping from substrates with conical geometric heterogeneity, and compared the results with those of planar substrates with different wettability and hydrophobicity. The results show that droplet dynamics can be enhanced by applying topographical heterogeneity. However, increasing the height of the cone does not always provide better conditions for jumping, and there is an optimal value for the height of the cone. This enhancement is due to the fact that more liquid flow affects the pressure gradient within the droplet, resulting in higher jump velocities. For flat surfaces, most of the kinetic energy can be converted into oscillations of the droplet during retraction and does not promote droplet jumping.

Although the research on Newtonian fluids has made great progress, the research on non-Newtonian fluids (viscoelastic fluids) is still very little. In the study of non-Newtonian fluids [35,36,37,38,39,40,41] most of the studies are on motion in microchannels. There is no study on the dynamics of viscoelastic droplets between a dielectric plate and on substrates of different wettability under the corona discharge emitted by a needle-plate electrode system. It has a diverse applicability due to its flexibility and maneuverability, and we examine through experiments with droplets of PEO aqueous solutions the entire process of droplet deformations by varying the PEO concentration, electric field strength, and surface wettability of the conducting substrate. Notably, we make use of a novel steerable system that uses a pin-plate electrode to polarize the dielectric to create a steerable electric field. We can adjust the position of the needle to make the electric field appear diagonally above the droplet, so that the lateral migration of the droplet can be achieved by the force of the electric field.

## 2. Materials and Methods

The experimental apparatus is composed of an iron needle of diameter 1 mm, a copper plate (10 cm × 10 cm × 1 mm), an acrylic plate (5 cm × 5 cm × 2 mm), an ITO conductive glass (3 cm × 3 cm × 2 mm), and a high voltage DC power supply, as shown in Figure 1. The distance between the needle and the acrylic plate is 1 mm (The horizontal position of the acrylic plate does not affect the movement of the water droplets. As the acrylic plate is a homogeneous dielectric, moving the acrylic plate horizontally will not affect the magnitude of the electric field, so it will not affect the movement of the droplet. The vertical position of the acrylic plate affects the movement of the water droplets, because the spacing of the needle from the acrylic plate affects the polarization of the acrylic plate and thus the magnitude of the electric field). The distance between the acrylic plate and the ITO plate was 5 mm. The droplets used in this experiment were all extracted by a pipette (Nichipet EX-Plus II pipette, volume 2–20 μL), and the volume was 5 μL. The experimental results are taken by a high-speed digital camera (MotionXtra N4), the lens is a macro lens (DG Macro Lens 105 mm), and the light source used is a high-power LED (Cyclops 1). During shooting, the total number of frames is 1912 frames, with the frame rate of 300 FPS and the shooting time of 6.37 s. The acrylic plate is irradiated by the needle-tip corona discharge, generating a local electric field. The electric field can be controlled by adjusting the position of the needle to control the movement of the droplet [42]. The ITO conductive glass plate supporting liquid droplets is placed above the copper plate. The surface treatment of the glass plate is performed to obtain different contact angles. The contact angle for the original untreated plate is 70°. With a thin PDMS layer of thickness less than 1 mm placed on the glass plate, the contact angle is measured as 105°. A superhydrophobic surface can be prepared by spraying a thin layer of hydrophobic nanoparticles (Glaco Mirror Coat Zero, Soft 99, Osaka, Japan) on silicon wafers to form loose and porous structures, so it has an ultra-low surface energy. Here, superhydrophobic surface was prepared by spraying with Glaco nanoparticles 1 and 3 times to obtain contact angles of 135 and 150 degrees, respectively. We prepared adhesives by dispersing 1 M poly(ethylene oxide) (PEO with average molecular weight $\mathrm{Mw}=1\;\times \;10^{6}{\;g\;\mathrm{mol}}^{-1}$, Sigma Aldrich, Burlington, MA, USA) in pure water (18.4 MΩ cm, Millipore Synergy, Darmstadt, Germany) elastic solution. To determine the effect of liquid viscoelasticity on droplet dynamics, we prepared a low and a high-concentration PEO aqueous solutions of 0.2% and 1%, respectively (at a molecular weight of $1\times 10^{4}{\;g\;\mathrm{mol}}^{-1}$, the solution at 2% concentration was non-viscous elastic gel, so 1% concentration of $\mathrm{Mw}=1\;\times \;10^{6}{\;g\;\mathrm{mol}}^{-1}$ molecular weight can be regarded as high concentration of PEO), and compared with the 0% pure aqueous solution.

## 3. Results and Discussions

Figure 2a shows droplet shapes on the ITO glass substrate before and after applying the electrical field in the pin-plate electrode system. The initial droplet shape, shown on the left, varies depending on the wettability of the substrate. For a droplet of volume 5 μL, the initial height $h_{0}$ is measured as $h_{0}$ = 1.3 mm, 1.8 mm, 2.08 mm and 2.12 mm, respectively, for contact angles 70°, 105°, 130° and 150°. After applying the voltage (11 kV), the droplet starts to deform soon after a time lag for the polarization of the top acrylic plate. For all three scenarios shown in Figure 2, evolutions to an equilibrium, to a bridge, and to a take-off are not monotonic, but with transient oscillations due to competing effects of gravitational, capillary, and electrostatic forces. Additional effects of viscoelasticity seem to amplify (for bridge and take-off) or weaken (for equilibrium) these oscillations depending on the scenario, but always shorten the evolution time. On hydrophilic substates droplets have relatively bigger footprints with lower height, and equilibration occurs after the oscillatory transiency with the balance of forces involved. On hydrophobic substrates droplets can touch the top acrylic plate, eliminating the possibility of equilibration. It is observed that regardless of the wettability of the substrate the time lag is longer for Newtonian droplets than those with nonzero PEO concentrations. The droplet deformation is accompanied by the change in its height from the initial value $h_{0}$ to $h_{t}$ with time. Figure 2b–d show the change in the droplet height against time in measures of its initial value:$$H=\frac{h_{t}}{h_{0}}$$

On the hydrophilic surface (contact angle of 70°) droplets are stretched to higher heights due to the electric field. For all three PEO concentrations transient small-amplitude oscillations are obvious before stationary equilibrium droplet shapes are reached. It is seen that both the equilibrium height and the time required for the equilibration decrease with the PEO concentration, whereas the time required for the equilibration. It can be speculated that on more hydrophobic substrate an equilibrated droplet can reach the top acrylic plate. On the superhydrophobic surface (contact angle of 150°) even more pronounced transient oscillations observed, followed by droplet take-offs from the substrate. The take-off time decreases with the PEO concentration. Viscoelastic droplets thus take shorter time for take-off, but with more complicated transiency of droplet bouncing. Upon take-off viscoelastic droplets also create a thin filament tail attached to the substrate. Neither the bouncing before take-off nor the tail after take-off is observed with water droplets. Figure 2e shows the total time after the application of the electric field required for a launching droplet from the superhydrophobic substrate to touch the top acrylic plate against the PEO concentration. The higher the elasticity due to increase in the PEO concentration, the shorter becomes the droplet to reach the top acrylic plate.

It is observed that on a hydrophilic substrate droplets tend to equilibrate to a final stationary state. For low voltage applied the equilibration is monotonic, whereas for high enough voltage rather severe transient vibrations exist before eventual equilibration, as shown in Figure 2b. Figure 3 shows the final equilibrium droplet height beyond the transient vibrations depending on the PEO concentration for an applied voltage of 10 kV. It is found that the final equilibrium height $H_{f}$ decreases with the increase in PEO concentration, or the increase in the elasticity of the droplet.

Figure 4 shows droplet evolution on a hydrophobic substrate with a contact angle of 105°, which is not high enough to exhibit the bridge with the top acrylic plate or the take off. As with the hydrophilic substrate, for the high enough voltage (11 kV) applied the droplet goes through severe transient oscillations before reaching the final equilibrium state. Both the maximum transient peak $H_{max}$ for the droplet height and the final equilibrium height are seen to decrease with the PEO concentration. The contact angle observed is insensitive to the PEO concentration tested in this study. The initial droplet height thus is independent of the PEO concentration. Regardless of PEO concentration droplets on a hydrophobic substrate would stand higher than those on a hydrophilic substrate. Figure 4b shows the transient maximum and final equilibrium droplet height with respect to the PEO concentration. Decrease in these heights with the PEO concentration is obvious, as seen also on a hydrophilic substrate. While the transient maximum is consistently higher on the hydrophobic surface, however, the equilibrium height observed is higher on the hydrophilic substrate. This reverse in the height is caused by the electrowetting present only with the dielectric PDMS substrate used for the contact angle $105^{{}^{\circ}}$ [43].

The phenomena shown above, including the severe transient oscillations before equilibration, bridging with top acrylic plate, and the take-off from the substrate, require high enough electric field to be applied. It is thus of great importance to identify the critical voltage for each phenomenon, which should vary with the wettability and the elasticity of the droplet. In Figure 5, the critical voltage is shown for three different contact angles and PEO concentrations for a droplet volume of 5 μL on the ITO glass. The severe transiency with final equilibrium shown on a hydrophilic substrate (contact angle 70°) is to be observed for voltages exceeding 10.9 kV, 10.6 kV, and 10.4 kV, respectively, for PEO concentration of 0%, 0.2%, and 1%. The critical voltage is seen to be maximum for the Newtonian droplet, and to decrease with the increase in the elasticity. Below these critical voltages, droplets evolve monotonically to an equilibrium without complicated transiency. The bridging phenomenon of droplet reaching the top acrylic plate is observed for voltages higher than 7.9 kV, 7.68 kV, and 7.6 kV, respectively, for PEO concentration of 0%, 0.2%, and 1%. Again, more elastic droplets require lower voltage for the bridging. Below the critical voltage, droplets saturate to an equilibrium, as shown in Figure 5. The critical voltage for droplets to take off from the superhydrophobic substrate also decreases with the PEO concentration, as shown for the case of 150° contact angle. For all cases shown, droplets reach an equilibrium shape for a low applied voltage. The critical voltage to escape this monotonic equilibrium decreases with the contact angle and the PEO concentration.

Figure 6 shows that at a contact angle of 130° the droplet is stretched to the top acrylic plate to form a liquid bridge. Here, Newtonian and elastic droplets exhibit different phenomena. After the water droplet touches the top acrylic plate, the resulting liquid bridge quickly disappears, and so the liquid filament is not sustained. The PEO aqueous solution on the other hand shows sustained liquid filament, more stable with higher PEO concentration. The liquid filament gradually becomes thinner, and reacts to the electric field. When the vertical electric field is weakened by the PEO aqueous solution attached to the top acrylic plate, the PEO liquid filament will be attracted by the nearby strong electric field because the top acrylic plate is uniformly polarized under the electric field. The liquid filament thus moves with a circular trajectory.

In the take-off phenomenon of droplets detaching from the superhydrophobic substrate, water and PEO aqueous solution also produced different behaviors. As shown in Figure 7, after the electric field (11 kV) is applied, the water droplet is stretched for a whole, and then the bottom is detached from the substrate, forming an oblate sphere, flying towards the top acrylic plate. The PEO aqueous solution takes a long time in the stretching process, and flies to the top acrylic plate in a spherical shape when separated from the wall, with a liquid filament appearing at the bottom during the flight. During the take-off process of the high-concentration PEO aqueous solution, the top is slightly stretched into a cone, and there is no obvious deformation in the process of flying to the top acrylic plate, but with a very thin liquid filament formed at the bottom.

For the process of droplet takeoff in Figure 8, comparisons are made for Newtonian and 0.2% PEO droplets. We compared the changes of position and velocity during the take-off of the droplet, respectively, using the previous time interval in which the severe deformation occurred as a reference target. We determined the position of the droplet by observing the value of H of the droplet, we found that the H value of the droplet before the violent deformation (initial position) was larger than that of the PEO droplet, and the bottom of the droplet collided with the top because the bottom of the droplet was detached from the bottom plate, this results in a faster drop rate for water droplets and a slower, more gradual drop for PEO droplets. In terms of speed, the water droplets fly much faster than PEO droplets, and touch the top acrylic plate in a shorter time. Negative velocities for a time interval are generated as the droplets collide and fuse in the air. Here, the recorded time interval is 33 ms.

The critical voltage for the take-off phenomenon of the droplet on the superhydrophobic (contact angle 150°) substrate is further investigated, as show in Figure 9. Three different droplet volumes of 5 μL, 10 μL, and 20 μL are chose to analyze the effect of droplet size along with PEO concentrations ranging from 0 to 1%. The take-off voltage required for water droplet (0%) is significantly higher than that of PEO aqueous solution, and the take-off voltage required by PEO aqueous solution has a smooth linear decay with the increase in concentration. As the concentration of PEO aqueous solution increases, the dielectric constant decreases gradually, and so the required voltage also decreases gradually. The critical take-off voltage increases with the droplet volume, regardless of the PEO concentration. Previous studies reveal that the practical contact area between a hydrophobic surface and water governs the net charge amount of the droplet on the surface [44,45]. A larger droplet thus increases the Coulomb force because it increases the electric charge on the droplet. A larger droplet size also increases the moving resistance because of the increase in the three-phase (solid–liquid–air) contact lines that are known to govern the movement of water droplets on solid surfaces [46]. Droplet size has more of an effect on Coulomb force than on the moving resistance, as the former depends on the contact area while the latter depends on the contact line. Therefore, the electric field required for droplet movement decreases with the increase in droplet size. Under a vertical electric field, Coulomb force on the droplet should overcome the sum of the adhesion force and gravity. Both adhesion force and gravity increase as the droplet size increases. The adhesion force is originally the same as the moving resistance mentioned above, and depends on the three-phase contact lines. However, gravity to a water droplet depends on the droplet mass, namely the droplet volume. The droplet size has more of an effect on gravity than on Coulomb force because the former depends on the droplet volume while the latter depends on the contact area, which relates to the square of the droplet radius [47]. The electric field required for droplet launching thus increases as the droplet size increases.

## 4. Concluding Remarks

The dynamics of viscoelastic droplets on surfaces with different wettability activated by a pin-plate electrode system with a dielectric plate is investigated for the first time. The viscoelasticity of droplets is controlled by changing the concentration of PEO in the aqueous solution. It is found that on hydrophilic and weakly hydrophobic surfaces (up to contact angle 105°), droplets under high electric field equilibrated to a stationary shape after severe transient oscillation. Regardless of wettability, droplets tend to equilibrate to a stationary shape under sufficiently low electric field. On highly hydrophobic substrates, droplets are stretched by the electric field to reach the top acrylic plate, and a bridge is formed. The liquid bridge formed by viscoelastic droplets show sustained filament in the middle, which thins and spins with the action of the electric field, which is analogous to an EHD electrospinning. Both water and PEO droplets take off from a substrate with even higher hydrophobicity. As in the bridging phenomenon, liquid filaments appear at the bottom of PEO droplets during the take-off process, in contrast to water droplets. The critical voltages for the phenomena reported in this work are also investigated with the viscoelasticity, the wettability, and the droplet size as parameters. We found that the voltage required for water take-off is higher than that of PEO aqueous solution, and the voltage required for take-off decreases linearly with the increase of PEO concentration. The voltage required for the droplet to take off increases with the droplet volume. The pin-plate electrode system is used to control the dielectric polarization to obtain the electric field. The motion of droplets can be controlled precisely by changing the position of the needle along with other parameters reported here. This work can be further extended to develop a new droplet manipulation method of practical importance.

## Figures and Tables

**Figure 1 micromachines-13-00580-f001:**
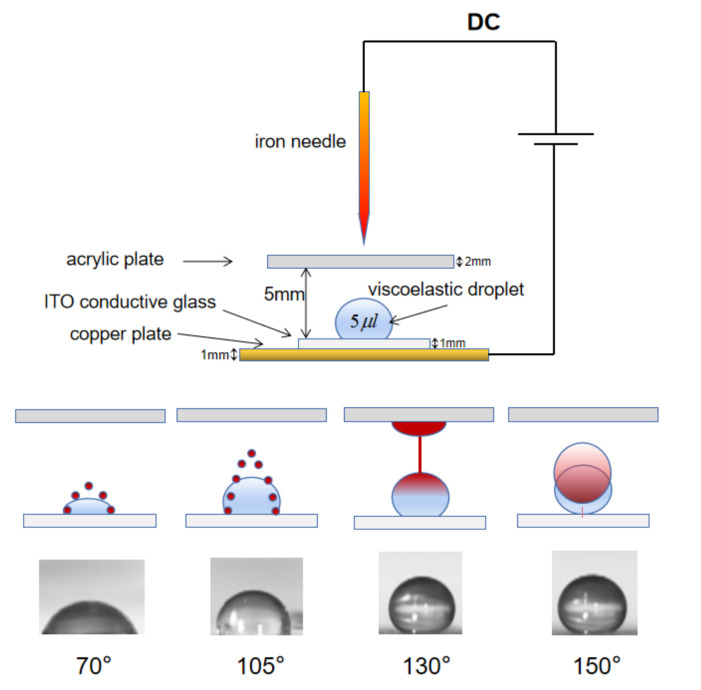
Deformation of viscoelastic droplets between a needle-plate electrode system. The electrodes are composed by an iron needle and a copper plate, and the acrylic plate is used as a platform to receive the corona discharge, and is polarized to generate an electric field, making the ITO glass conduct electricity and droplets on it deform. Behaviors of droplets for different contact angles of 70, 105, 130, and 150 degrees. The droplet deformation reaches an equilibrium for contact angles of 70 and 105 degrees, whereas for 130 degrees the droplet is stretched to the top acrylic plate, forming a liquid bridge a thin filament in the middle. For 150 degrees, the droplet can take off toward the top acrylic plate.

**Figure 2 micromachines-13-00580-f002:**
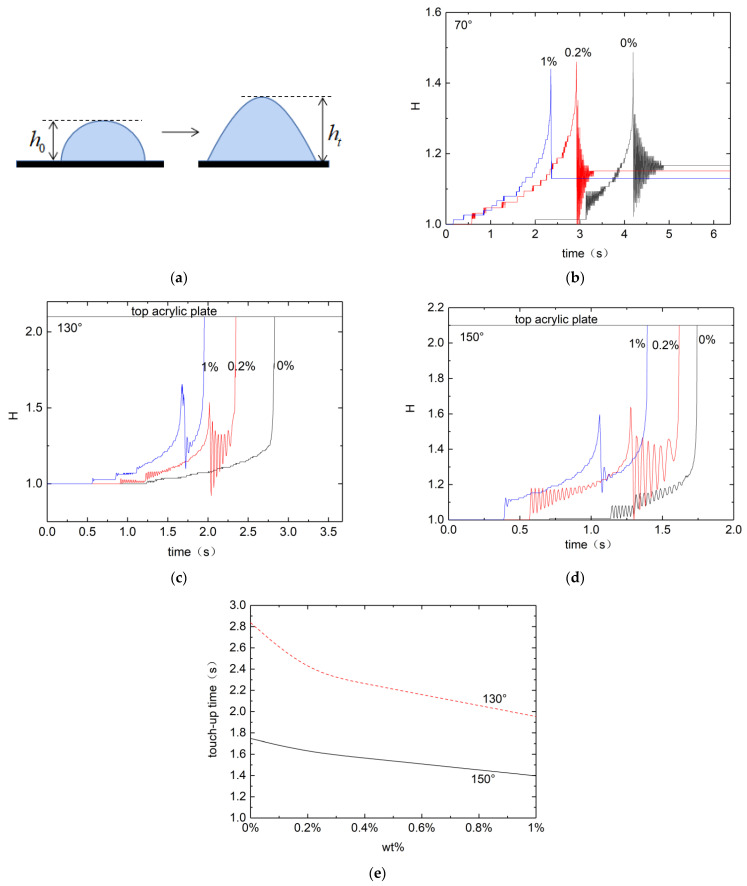
Deformation process of water, low-concentration PEO aqueous solution, and high-concentration PEO aqueous solution at different contact angles. (**a**) Initial and deformed droplet configuration. (**b**–**d**) Dimensionless droplet height evolution for contact angle of 70°, 130° and 150°. (**e**) Dimensionless time required for droplet tip to reach the top acrylic plate against PEO concentration. Voltage applied through the pin-plate electrode system is 11 kV.

**Figure 3 micromachines-13-00580-f003:**
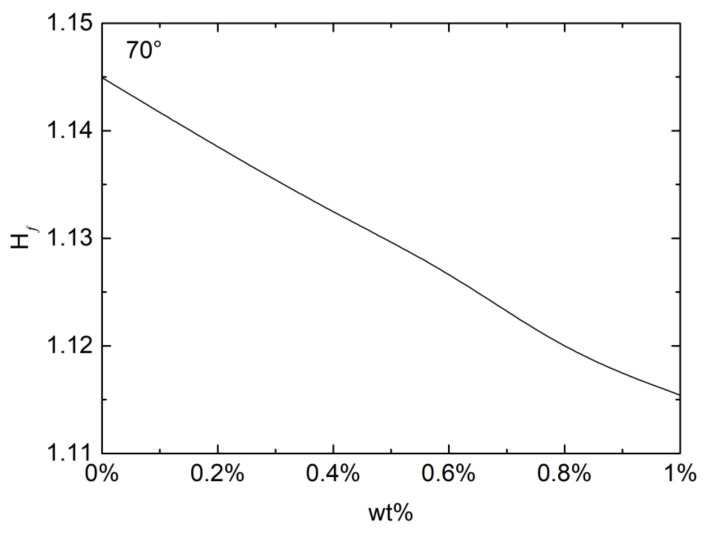
Equilibrium droplet height with respect to PEO concentration at a voltage of 10 kV and a contact angle of 70°.

**Figure 4 micromachines-13-00580-f004:**
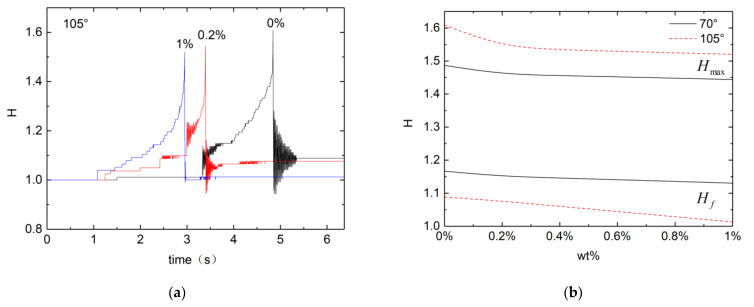
Droplet deformation on a dielectric PDMS substrate with contact angle 105°. (**a**) Droplet height against time; and (**b**) equilibrium height against PEO concentration.

**Figure 5 micromachines-13-00580-f005:**
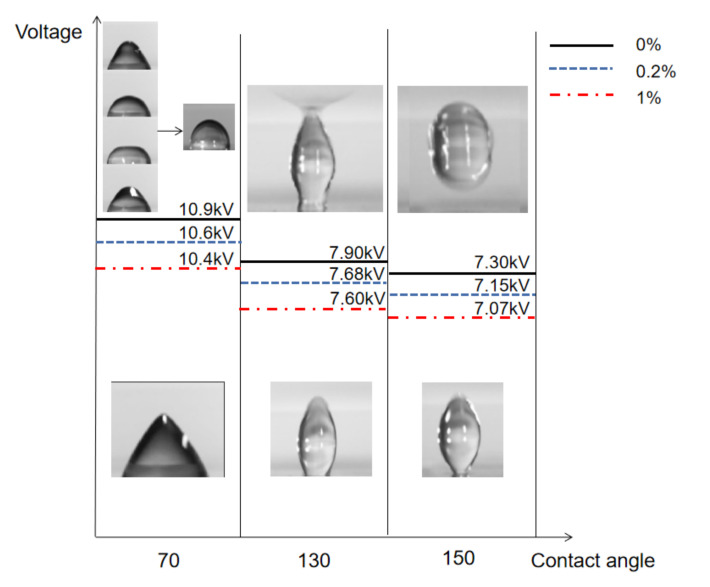
Critical voltage to overcome monotonic equilibration to a static droplet on hydrophilic, hydrophobic, and superhydrophobic substrate for three different PEO concentrations.

**Figure 6 micromachines-13-00580-f006:**
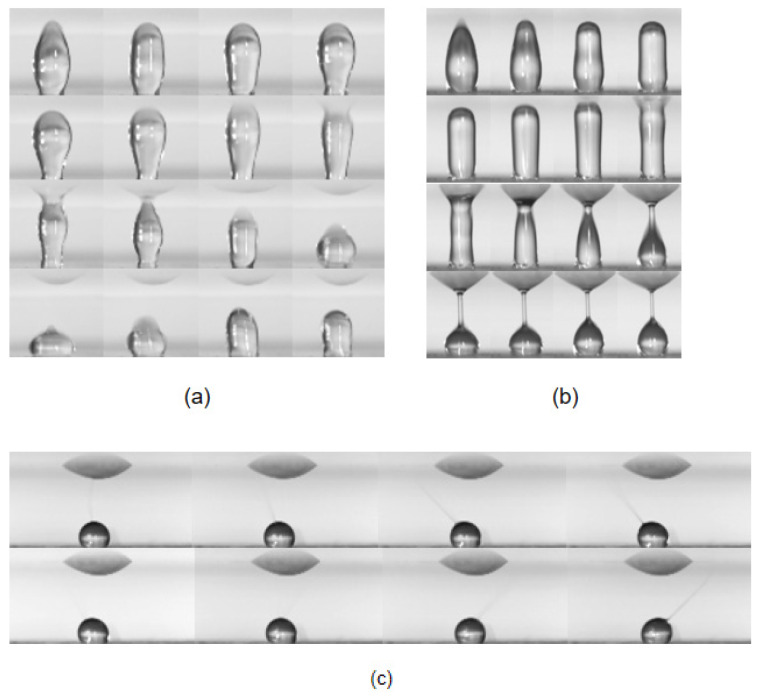
The liquid bridge phenomenon at a contact angle of 130°. (**a**) After the water contacts the top acrylic plate, a liquid bridge is formed for a short time, and then disappears; (**b**) PEO aqueous solution produced obvious liquid filaments after contacting the top acrylic plate; and (**c**) a phenomenon similar to electrospinning appears after the liquid filaments becomes thinner.

**Figure 7 micromachines-13-00580-f007:**
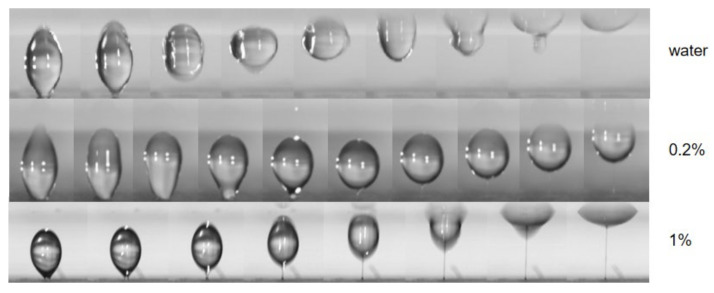
The sequence of water, 0.2% PEO aqueous solution and 1% PEO aqueous solution droplet taking off from superhydrophobic substrate. The water droplet shows no filament formation, while PEO droplets do.

**Figure 8 micromachines-13-00580-f008:**
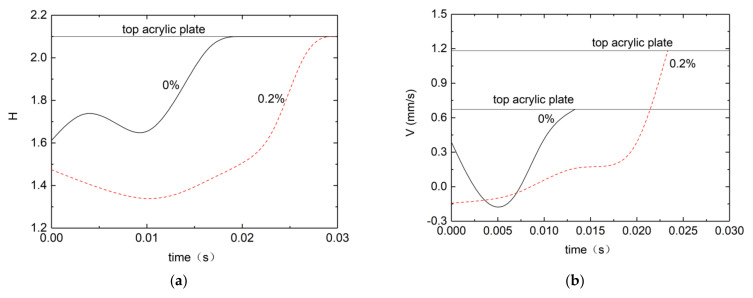
Droplet position and velocity vs. time for Newtonian and viscoelastic droplets. (**a**) Position of the droplet tip. (**b**) Velocity at the droplet tip for water and 0.2% PEO aqueous solution taking off on a superhydrophobic surface (150°).

**Figure 9 micromachines-13-00580-f009:**
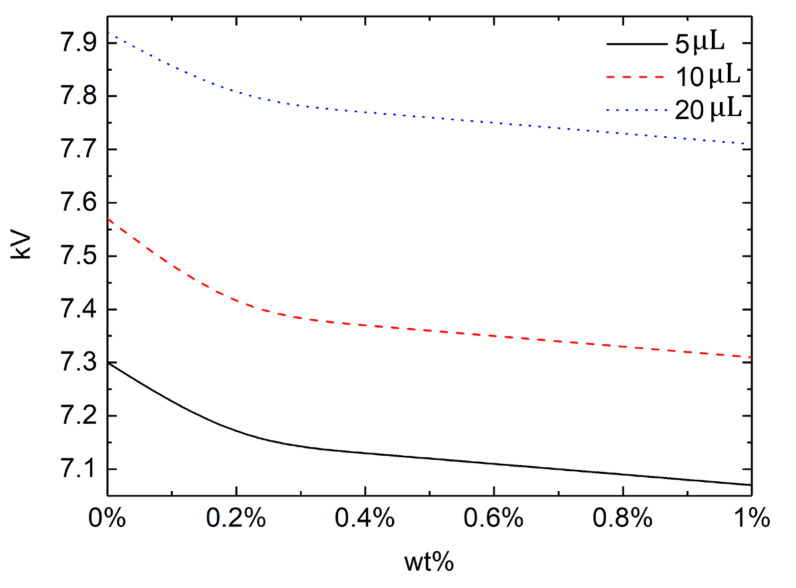
Critical voltage for the take-off phenomenon against PEO concentration for three different droplet volumes.

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
