# Peer review of "The Effect of Surface Wettability on Viscoelastic Droplet Dynamics under Electric Fields"

_micromachines, 2022, doi:10.3390/mi13040580_

Round 1
Reviewer 1 Report
Dear Authors,
I understand you put a lot of effort in performing the research and drafting the manuscript. However, I find the manuscript to be written in a sloppy way. There are a lot of language and style errors. Figures need editing--there are too small, have too small fonts, and in some figures certain features are not described, so that it is hard to tell what the figures represent.
These scattered a lot of my attention while reading and I had problems to properly evaluate the scientific content of your work.
Once the data presentation is improved, and all mistakes are corrected, I will be glad to revise your manuscript from the point of view of scientific merit.
For a moment, I do not recommend publishing the manuscript in this form.

Author Response
我们非常感谢您的宝贵建议。

Reviewer 2 Report
The authors have performed an experimental study of the effect of an electric field (range of 10 kV) on the shape and the transport of Newtonian an viscoelastic fluid drops in a narrow space between a polimeric screen and a conductive ITO surface. They discuss the change of shape from hemispherical to a quasi-conical, and the deformation to form a liquid bridge or to present bouncing between the plates. Although the results are interesting, the physical phenomena is not new at all. Some aspects of the manuscript have to be addressed before it can be accepted for publication.
COMMENTS:
1.- In page 2 kappa1 and kappa2 have to be explicitly defined despite the reader may guess its meaning.
2.- The size of the symbols in the equations inside the text should be equal to the letters of the text.
3.- Page 4: Although the voltage used is indicated in the legends of the figures, it must also appear in the text where the figures are mentioned.
4.- The authors have to give a clear and detailed explanation of the origin and meaning of the transients in Figure 2.
5.- The introduction contains some possible applications of the phenomena studied. However, it is not discussed how the present results can help to improve the ability to predict or improve the design of such applications.
6.- The authors mention in different parts of the manuscript that there is competence between gravity, change of surface tension and electrical force. However, they do not quantify those three factors, e.g., how much the surface tension changes when an electrical potential of aprox. 10 kV is applied? Is there any temperature increase in the liquid when the field is applied? Can one disregard any field gradient between the cusp and the bottom of the drop? If any gradient exists a Marangoni stress will exist due to the field dependence of the surface tension.
Author Response
We gratefully appreciate for your valuable suggestion.

Reviewer 3 Report
The paper presents experimental results on the vertical displacement/motion of a viscoelastic droplet under the influence of an externally applied electric field. The results are interesting and should be of interest to the readers of the journal. However, sufficient details about the experimental condition and/or data acquisition methods have not been properly reported. The quality of the figures also needs to be improved. The reference section also needs to be improved. After these modifications are made, the paper can be considered for publication.
-
The relevant dimensions should be shown in Figure 1. For example, the diameter of the droplet, separation between the acrylic sheet and the ITO plate, thickness of the plates (ITO, copper, acrylic) etc. In case an exact number is not available, please put in a range of values. These information will improve the readability of Figure 1 significantly.
-
Please put units on the x axis of Figure 2(b)-(e). If the y axis is unit-less, please mention that on the axis label.
-
The three lines/plots in Figure 2(b) should be plotted in different colors and with different markers. It is hard to separate out the three plots as they are all in black without any markers. Please do this for Figure 2(c), (d) and (e) as well.
-
From Figure 2(c) and (d), it appears that the plots have not reached any kind of steady state value. It the top of the plot indicates the droplet touching the acrylic glass, please label that on the plot.
-
Please explain in a bit more detail why the hydrophilic substrate droplets tend to find an equilibrium position.
-
Please mention the imaging setup used to capture/measure droplet height data. The camera model, lens configuration/model, lighting condition, frame integration time, and the frame rate should be mentioned. Any other relevant information should also be included.
-
When comparing droplets of different sizes, the authors have briefly discussed the moving resistance and the Coulomb force. This discussion should be more elaborate. Also, would gravity have any effect in this? What about surface tension? Please discuss.
-
The magnitude of the electric field has not been mentioned anywhere in the paper. As the motion is driven by the electric field (which depends on the voltage as well as the separation between the electrodes), it should be explicitly mentioned.
-
In the introduction section, the authors mention several techniques of driving motion in microfluidic devices. Perhaps a few papers on advanced dielectrophoresis methods (e.g. traveling wave dielectrophoresis, moving dielectrophoresis) can be cited:
-
https://doi.org/10.1002/elps.1150130110
-
https://doi.org/10.1063/5.0049126
-
-
Optoelectronic tweezers technology are also being used in microfluidic device. It should be briefly mentioned with a recent references (e.g. Prof. Ming Wu and Prof. Lambertus Hesselink’s works on the topic)
-
Would it be possible to discuss some mathematical modeling of the observed phenomenon?
-
Could similar phenomenon be observed in the horizontal direction? If so, then the droplet can be transported along a horizontal direction which can be of interest to the microfluidic community. Similar electro-wetting phoenomenon have been reported. But perhaps, a viscoelastic droplet transportation can be rather interesting. The authors may pursue such ideas in future. A brief statement on this on the current paper would be good.
Author Response

(The authors gave the same response as above.)

Round 2
Reviewer 1 Report
The authors have improved the quality of the data presentation that meets now the journal standards.
Reviewer 3 Report
All the suggested changes have been incorporated. The paper can be considered for publication now.